# Current Controversies on the Pathogenesis of Medication-Related Osteonecrosis of the Jaw

**DOI:** 10.3390/dj4040038

**Published:** 2016-10-28

**Authors:** Winnie Zee Man Wat

**Affiliations:** Department of Medicine, Pamela Youde Nethersole Eastern Hospital, Hong Kong, China; watzm@ha.org.hk; Tel.: +852-25956111

**Keywords:** osteonecrosis of the jaw, BRONJ, MRONJ, DRONJ, bisphosphonate, zoledronate, denosumab, anti-angiogenic

## Abstract

Medication-related osteonecrosis of the jaw (MRONJ) was first reported more than a decade ago. Since then, numerous cases have been diagnosed. Currently, there are three groups of drugs related to MRONJ: bisphosphonates, denosumab and anti-angiogenic drugs. As MRONJ can lead to debilitating clinical sequels and limited effective treatment options are available, much research has been done in understanding its pathophysiology. Until now, the exact pathogenesis of MRONJ has not been fully elucidated. While history of invasive dental procedures or local trauma may be present, some cases occur spontaneously without any preceding factors. This review aims to examine and discuss the three main hypotheses for the pathogenesis of MRONJ, namely suppressed bone turnover, cellular toxicity and infection.

## 1. Introduction

In 2003, Marx et al. reported 36 cases of osteonecrosis of the jaw (ONJ) related to the use of either zoledronate or pamidronate. They were presented with painful bone exposure in the mandible, maxilla, or both, and were unresponsive to surgical or medical treatment [1]. Since then, numerous similar cases of bisphosphonate-related ONJ have been reported in the literature. 

In 2014, the term “medication-related osteonecrosis of the jaw” (MRONJ) was proposed by the American Association of Oral and Maxillofacial Surgeons to describe ONJ related to the use of medications [2]. To diagnose MRONJ, all the following four criteria should be fulfilled: (1) current or previous treatment of bisphosphonates or medications known to be related to ONJ; (2) bone in the maxillofacial region exposed persistently for more than eight weeks; (3) absent history of radiation to the jaw; and (4) absence of malignancy in the involved region [3,4,5]. 

The clinical presentations of MRONJ vary widely. Many cases are presented with poor wound healing or spontaneous intra-oral mucosal breakdown that subsequently leads to the exposure of the underlying bony structure and finally bone necrosis in the oral and maxillofacial region [6,7]. While history of invasive dental procedures or local trauma may be present [8,9,10], some are presented as incidental findings of exposed alveolar bone during routine dental assessment [11]. 

The majority of MRONJ involves the mandible, and bony prominences with thin mucosal coverings are common vulnerable sites. It is estimated that 65% of cases are located in the mandible, 28.4% in maxilla, 6.5% in both mandible and maxilla, and 0.1% in other locations [8,12,13]. Several postulations have been proposed for the exclusive localization of osteonecrosis at the jaw bones. Local factors likely play significant roles. Dentoalveolar surgery has been identified as a major risk factor for MRONJ, with 52% to 61% of patients reporting tooth extraction as the precipitating event [14,15,16]. In cancer patients with exposure to zoledronate, tooth extraction is associated with a 16- to 33-fold increased risk of MRONJ [14,17]. The risk for developing ONJ after tooth extraction in patients with cancer exposed to intravenous bisphosphonates ranges from 1.6% to 14.8%, which is much higher than the placebo group in clinical trials that ranges from 0% to 0.019% [2].

As prevention remains the mainstay of management for MRONJ, an in-depth look into the pathophysiology of the disease may shed some light on how to reduce its occurrence. At present, the exact pathogenesis of MRONJ has not been fully revealed. This review aims to look into the existing potential mechanisms of MRONJ, namely suppressed bone turnover, cellular toxicity and infection. Currently, there are three groups of drugs related to MRONJ including bisphosphonates, denosumab and anti-angiogenic drugs. Different terms have been used for specific drug groups: bisphosphonates-related ONJ (BRONJ) and Denosumab-related ONJ (DRONJ) [18]. Though the proposed pathogenesis for BRONJ and DRONJ are similar, each drug group will be discussed separately in this review as their mechanisms of action are different (Figure 1). 

## 2. Bisphosphonates

Bisphosphonates are analogues of naturally occurring inorganic pyrophosphates. With the replacement of the oxygen atom by the carbon atom in the pyrophosphate backbone that connects the two phosphates, they are resistant to natural chemical and biological degradation. 

Currently there are three main hypotheses for the pathogenesis of BRONJ, namely suppressed bone turnover, cellular toxicity and infection. 

**(a) Suppressed Bone Turnover**

Suppressed bone turnover has been a popular hypothesis for the pathogenesis of BRONJ.

It is believed that reduced bone resorption followed by a reduction in bone-forming activity can lead to the accumulation of microdamage and an area of necrosis [19]. This theory has been supported by an animal study showing areas of necrosis developed in the mandibles of dogs treated with high doses of bisphosphonates [20], and case reports demonstrating healing of ONJ with the bone-forming agent teriparatide [21,22,23]. Besides, anti-resorptive agents may provide a stable and unresponsive surface for microbial colonization and development of secondary infection in the presence of minor mucosal trauma or dental extraction [24,25].

In line with this hypothesis, epidemiology data has revealed that more potent bisphosphonates, more frequent dosing, a higher accumulative dose and a longer duration of treatment are all major risk factors for BRONJ. In a survey of 13,000 subjects on oral bisphosphonate (alendronate), the overall incidence of BRONJ rose from 0.1% to 0.21% in those on alendronate for more than four years [26]. A similar trend has also been observed in cancer patients receiving zoledronate, with the incidence of ONJ being 0.6% with one year’s treatment of zoledronate, 0.9% with two years’, and 1.3% with three years’ [15]. 

In spite of these observations, there are studies showing accelerated rather than suppressed bone turnover at the ONJ lesions. Significant tracer uptake in the bone scan and increased numbers of osteoclasts and bone lysis have been found at the lesions [27,28]. Moreover, patients with osteopetrosis whose osteoclasts are absent or nonfunctional do not develop ONJ as a disease complication, and there is no evidence of spontaneous bone necrosis at other skeletal sites with bisphosphonates and denosumab. In addition, denosumab being a more potent anti-resorptive agent has the incidence of ONJ comparable to or only slightly higher than zoledronate [29]. In the combined data of two studies consisting of around 2000 patients receiving either denosumab or zoledronate for the treatment of malignancy-related bone metastasis, the incidence of ONJ was 1.5% and 1.3%, respectively [30]. The overall incidence of ONJ in cancer patients receiving denosumab was reported to be 1.7% in a recent meta-analysis that included a total of 8963 patients with a variety of solid tumors in seven randomized controlled trials [31]. With these arguments, suppressed bone turnover may not be the primary pathophysiology of ONJ in spite of the apparent drug action of bisphosphonates. 

**(b) Direct Cellular Toxic Effects**

Some investigators have proposed BRONJ may be caused by the direct toxic effects of bisphosphonates to the epithelial cells and macrophages, which disrupts the integrity of the oral mucosa and leads to infection and bone necrosis. 

It is known that nitrogen-containing bisphosphonates affect the function and survival of the osteoclasts by inhibiting the farnesyl pyrophosphate synthase enzyme of the mevalonate pathway in cholesterol synthesis. Zoledronate is the most potent inhibitor among all bisphosphonates. 

It has been shown that inhibition of farnesyl pyrophosphate synthase could cause toxicity to gastrointestinal epithelia [32], and bisphosphonate pretreatment of murine oral mucosal cells inhibited proliferation and wound healing [33]. It has also been found that the growth of a variety of cell types cultured on bone surfaces previously treated with bisphosphonates were inhibited in vitro [34]. Such toxic effects may compromise the healing of soft tissue after trauma or tooth extraction. While animal and in vitro studies showed that zoledronate inhibited the proliferation of human endothelial cells [35] and caused reduction in the revascularization of the prostate gland in rats [36], the fact that most reported BRONJ samples contain patent vessels [37,38] renders the role of ischemic necrosis in the pathogenesis of BRONJ likely minor.

As mentioned, bisphosphonates also have toxic effects on macrophages. It has been demonstrated that cholesterol synthesis would be inhibited [39] and apoptosis would be induced [40] after bisphosphonates were internalized into J774 macrophage-like cells [41]. It is thought that high-dose bisphosphonates used in cancer patients may reduce local immune response through this mechanism, and facilitate infection and bone necrosis in the presence of mucosal lining disruption.

**(c) Infection**

While infection is widely accepted as a contributing factor for MRONJ, its exact role as being the trigger or the sequel of the process is still unclear. 

It is believed that infection may contribute to ONJ by inducing excessive bone resorption. Patients with a history of inflammatory dental diseases such as periodontal and dental abscess are at seven-fold risk of developing BRONJ [42]. Some bacteria, especially Gram-negative species, produce lipopolysaccharides that stimulate local cytokine production and cause accelerated bone resorption [43,44]. Other bacteria have also been shown to directly regulate the production of receptor activator of nuclear factors-kB ligand (RANKL) in human periodontal ligament cells, gingival fibroblasts [45,46] and B cells, which can accelerate bone resorption [47]. 

## 3. Denosumab 

Denosumab is a human monoclonal immunoglobulin antibody [48,49] that inhibits the binding of receptor activator of nuclear factors-kB ligand (RANKL) in the cell membrane of osteoblasts to the receptor activator of nuclear factor-kB (RANK) in the cell membranes of osteoclasts and osteoclast precursor cells (Figure 1). As a result, it inhibits the cascade that leads to fusion of osteoclast precursors to form osteoclasts, attachment of osteoclasts to the bone, activation of osteoclasts, and delayed apoptosis of osteoclasts [50,51,52]. 

As in the case of bisphosphonates, many investigators postulated that denosumab induced osteonecrosis of the jaw via its anti-resorptive action, though little physiological and histological evidence is available to support such speculation. In contrast, the hypothesis of cellular toxicity has been actively explored. RANKL has several regulatory functions [53,54]. It is produced not only by osteoblasts but also activated T cells and keratinocytes [55], and its receptors are expressed on osteoclasts, monocytes, macrophages and dendritic cells. As RANKL stimulates the production of pro-inflammatory cytokines, reduces monocyte apoptosis and promotes the migration of these cell types, blockade of the RANK-RANKL interaction may affect monocyte and macrophage functions and survival [53] and increase the risk of infection and bone necrosis.

## 4. Anti-Angiogenic Drugs

Anti-angiogenic drugs are only lately noted to cause ONJ. The main action of this drug group is to inhibit the vascular endothelial growth factors (VEGF) that are over-expressed in most solid cancers [56,57,58], and thus suppress the neo-angiogenesis of the tumors. VEGF plays an important role in the regulation of osteoclastic function, and in the promotion of osteoclast differentiation and survival [59,60,61]. This group of drugs includes bevacizumab, sunitinib and cabonzantinib.

Bevacizumab is a humanized monoclonal antibody that blocks VEGF. It is associated with ONJ when given alone or with bisphosphonates. The reported ONJ incidence in a large study with 3560 subjects was 0.2% for bevacizumab alone, and it increased to 2.4% for combination treatment with bisphosphonates [62]. In a study, up to 70% of patients receiving combination treatment developed ONJ, and the onset time was much shorter with the combination treatment than with bevacizumab alone, namely 12.4 months and 23 months, respectively [63]. 

Sunitinib is an oral tyrosine kinase inhibitor which blocks multiple downstream cascade products including VEGF receptors. The risk of ONJ increases in patients receiving both sunitinib and bisphosphonates. In a small series where 21 patients with metastatic renal cell carcinoma were treated with both zoledronate and sunitinib, five patients (24%) developed ONJ after a mean duration of 18.5 months of zoledronate and 5.4 months of sunitinib [64]. Another study reported an incidence of ONJ in 10% of patients receiving concomitant bisphosphonate treatment [65].

Cabozantinib is another oral tyrosine kinase inhibitor [66] with action against VEGF for treatment of local advanced or metastatic medullary thyroid carcinoma. The incidence of ONJ was 1.4% in a phase III trial of the drug [67].

Currently, minimal data has been published on the pathogenesis of this group of MRONJ. Further studies are warranted to elucidate the underlying mechanism though inhibited angiogenesis or ischemia leading to bone necrosis is widely speculated.

## 5. Summary

Up until now, the pathogenesis of MRONJ has not been well clarified in spite of numerous cases being diagnosed and the groups of related medications expanding. Tooth extraction and invasive dental procedures are common precipitating events, though their presence is not a prerequisite for the development of MRONJ. Currently there are three main hypotheses proposed for the pathogenesis of MRONJ: suppressed bone turnover, cellular toxicity and infection. While circumstantial data supports the theory that anti-resorption reduces bone formation and promotes necrosis, physiological and radiological studies have shown conflicting findings. In contrast, the postulation of drug-related cellular toxicity to epithelial cells and macrophages resulting in impaired functions and apoptosis, and subsequent disruption of mucosal integrity and compromised immune defense has been supported by multiple tissue and cellular studies. Finally, infection being commonly present in MRONJ is believed to be a significant contributing factor to the disease as it can induce excessive bone resorption. Whetherit is the trigger or the result of the disease process remains unclear. Further studies are warranted to confirm the exact pathogensis of anti-angiogenic drugs causing MRONJ as their drug actions are different from bisphosphonates and denosumab.. Figure 2 summarizes the current perspectives on the pathogenesis of MRONJ.

With the uncertainty on the exact pathophysiology of MRONJ, it has been controversial whether a drug holiday should be recommended to patients receiving bisphosphonates, denosumab or antiangiogenic therapy who require tooth extractions. Currently, there is limited data supporting that the interruption of related drugs can alter the risk of ONJ in these patients undergoing tooth extraction or invasive dental procedures [68]. Proper oral hygiene, vigilant awareness and preventive dental care remain crucial to prevent MRONJ in high-risk patients. 

Finally, it is worthwhile to aware that existing evidence has shown that prolonged use of bisphosphonates for osteoporosis could only achieve a sustained reduction in vertebral fractures, but not non-vertebral fractures. Hence, it is advised by some experts to consider a drug holiday after five years of oral or three years of intravenous bisphosphonate treatment in low-risk patients as the risk of atypical femoral fracture, in addition to MRONJ, increases with the duration of bisphosphonate therapy [69]. Balancing the risks and benefits of treatment in individual cases remains the cornerstone of management in this controversial area, and regular update is mandatory.

## Figures and Tables

**Figure 1 dentistry-04-00038-f001:**
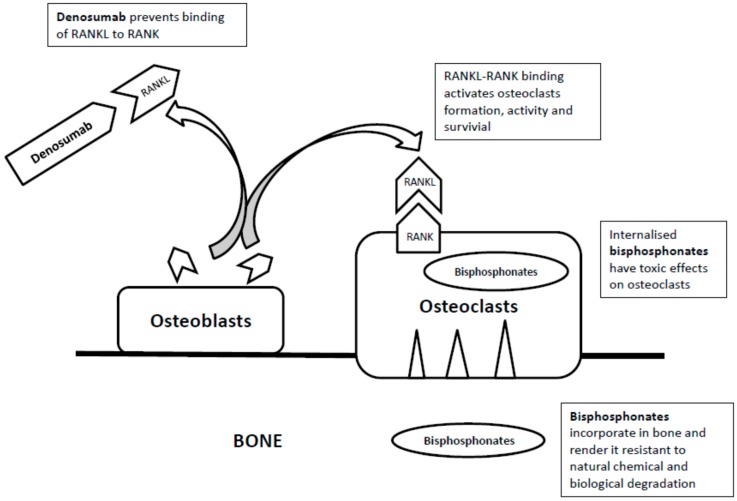
The mechanisms of action of bisphosphonates and denosumab.

**Figure 2 dentistry-04-00038-f002:**
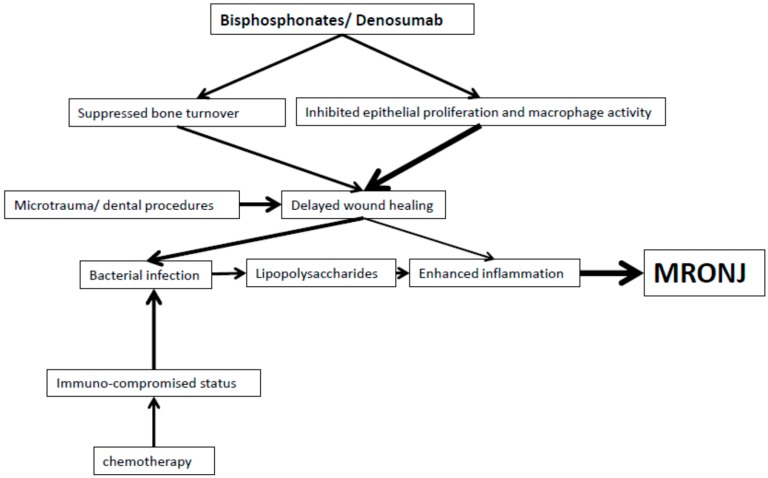
Flowchart shows the current hypothesis of the pathophysiology of MRONJ.

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
