# Peer review of "Current Controversies on the Pathogenesis of Medication-Related Osteonecrosis of the Jaw"

_dentistry, 2016, doi:10.3390/dj4040038_

Round 1

Reviewer 1 Report

This paper aims tp analyze the principal pathogenetic theories of MRONJ. The article offers interesting data for dentists, oral pathologists and maxilla-facial surgeon.

Unfortunately, some considerations are needed for your manuscript.

The analysis of the pathogenetic theories is filled of some redundant epidemiological data: you have to decide if change the title (including epidemiological aspects about ONJ) or to modify the structure of paragrapghs in which too much is expressed on ONJ epidemiology.

Furthermore, the use of the terms “MRONJ” and “BRONJ”, in some passages, is confused and needs to be revised. 

The discussion should include some new perspectives and/or proposal as conclusive arguments.

Overall, your article needs also the revision by  native English language specialist .

Author Response

Thank you for reviewing and feedback.  Relevant revision has been made accordingly.

Redundant epidemiological data has been reduced to keep in line with focus of the discussion.

The terms MRONJ, DRONJ and BRONJ have been defined and used accordingly if specific drug is discussed.

A short discussion on the recommendation of drug holiday related to MRONJ has been added in the summary section to enhance the clinical implication of this paper.  

English grammar has been extensively revised.

Reviewer 2 Report

Dear Author,

I welcome this paper and congratulate you on a good overall summary of this topic.

Below are some relatively minor points I suggest be addressed.

The overall grammar and punctuation should be reviewed because there were a small number of minor errors or passages that simply did not read very well. For instance, there was one sentence that started with "And". 

Two more examples:

"prostate gland in rates" presumably should read rats

and

"delay osteoclast apoptosis" should read delayed?

The terms MRONJ, BRONJ and ONJ are used inconsistently and interchangeably. I presume that this is because in each sentence you are referring to the particular paper and its term. While it may be difficult to explain this in each case, I suggest that you include a brief paragraph explaining where the term MRONJ came from, namely from the 2014 AAOMS Position Paper in the JOMS (Ruggiero et al). It would also be beneficial to mention DONJ as explained in O’Halloran et al (Aust Dent J. 2014). Please also read this paper and the papers referenced with regards to the incidence of ONJ (or more correctly DONJ) with denosumab (Prolia = 60mg sc every six months - for osteoporosis / prevention of). Please have a look at your sentence regarding the incidence of ONJ in these patients and look at the latest literature on this topic.

A few more minor points:

It would be good to further emphasise that there are increased incidences of ONJ with extractions and even trauma versus spontaneous occurrences. This may be helped by adding some supporting data / figures on this.

"Hence, the incidence of BRONJ in cancer patients is around 100 to 1000 times of that in osteoporosis patients. It is speculated that such high discrepancy in incidence may be related to the frequent dosing of zoledronate which is 4mg monthly in cancer patients, versus 5mg yearly in osteoporotic patients."

I feel that giving these relative dosages is misleading because they are iv versus oral, therefore cannot be directly compared. If you are to leave them in, please explain how they are not comparable.

If you wanted to include a diagram with this paper, a picture showing where the bisphosphonates work relative to where denosumab works may be interesting.

Finally, you state that "Denosumab has a relatively short half‐  life, 25 to 29 days". It may be more clinically relevant to discuss how its effectiveness is in the range of six months, hence the six monthly 60mg sc injections for osteoporosis / prevention of. This has been demonstrated with CTX levels.

Finally, it may be worthwhile including how new guidance (for instance from the FDA in the USA) highlight how it may not be beneficial for patients to continue bisphosphonates for periods greater than five years, in terms of increased fracture prevention. While your paper is on the pathogenesis, it is useful to put this disease process in context and mention related controversies that may influence the bigger picture of this disease in the  future.

Yours sincerely

Author Response

Thank you for reviewing and feedback.  Relevant revision has been made accordingly.

English grammar has been extensively revised.

The terms MRONJ, DRONJ and BRONJ have been defined and used accordingly if specific drug is discussed.

Redundant epidemiological data has been reduced to focus on relevant discussion.

Data on increase incidence of MRONJ with tooth extraction has been added.

A diagram showing the mechanism of action of bisphophosphate and denosumab in the bone metabolism is added.  A diagram summarises the discussed pathogenesis of BRONJ and DRONJ has been added.

A short discussion on the recommendation of drug holiday related to MRONJ has been added in the summary section to enhance the clinical implication of this paper.  

Round 2

Reviewer 1 Report

Please, your paper needs a revision by an English mother tongue. Now many mistakes are present.

Many tks for your consideration

Author Response

English revised.

Reviewer 2 Report

Dear Author,

There are still some points to address.

The first sentence of the introduction still causes confusion (as was pointed out by the other reviewer). "Medication related osteonecrosis of the jaw (MRONJ) was first reported by Marx et al in 2003 [1]." Marx did not name this MRONJ in 2003 so this could be viewed as inaccurate. This needs editing or the terms MRONJ, BRONJ and ONJ need explaining at this point. I would however, advise against making this first paragraph too complex with these explanations so early (you have done this well later on)! The term MRONJ came from, namely from the 2014 AAOMS Position Paper in the JOMS (Ruggiero et al). It would also be beneficial to reference this and also DONJ as explained in O’Halloran et al (Aust Dent J. 2014).

Instead of "up till now" being repeated over and over, I suggest that "until now" or other 'less slang' variations are used. The overall grammar and punctuation should be reviewed by a native English speaker / publisher.

Some more minor points:

It would be good to further emphasise that there are increased incidences of ONJ with extractions and even trauma versus spontaneous occurrences. 

Importantly, I suggest that you edit your point:

" Currently, there is no data supporting the interruption of related drugs can alter the risk of ONJ in these patients undergoing tooth extraction or invasive dental procedures. In spite of that"...

Please see the below 2014 paper I have copied below (J Oral Maxillofac Surg 72:1456-1462, 2014).

This has demonstrated that CTX levels may be of relevance. As I mentioned before and also below, you should consider including this clinically relevant information.

To ensure that your paper is clinically interesting to readers, it may be worthwhile including how new guidance (for instance from the FDA in the USA) highlight how it may not be beneficial for patients to continue bisphosphonates for periods greater than five years, in terms of increased fracture prevention. While your paper is on the pathogenesis, it is useful to put this disease process in context and mention related controversies that may influence the bigger picture of this disease in the  future.

Yours sincerely

A C-Terminal Crosslinking Telopeptide

Test–Based Protocol for Patients on Oral

Bisphosphonates Requiring Extraction: A

Prospective Single-Center Controlled Study

April Hutcheson,Andrew Cheng, MBBS, BDS, FRACDS(OMS),y

Ranjit Kunchar, MBBS, BDS,Brian Stein, MBBS, FRACP,x

Paul Sambrook, MBBS, MDS, FRACDS(OMS),k

and Alastair Goss, DDSc, FRACDS(OMS){

Purpose: Patients undergoing extraction are at risk for bisphosphonate-related osteonecrosis of the jaws

(BRONJ). A C-terminal crosslinking telopeptide (CTX) level lower than 150 pg/mL has been suggested as a

predictor of BRONJ risk. The authors aimed to increase the precision of estimates of the risk of BRONJ in

osteoporosis after extraction and to assess value of CTX testing at extraction time in cases of BRONJ in a

large prospective cohort.

Patients and Methods: All patients on oral bisphosphonates for osteoporosis referred for extractions

over a period of 6.5 years were included in a standard protocol. Pre-extraction fasted CTX levels were obtained.

All patients were followed until healing. If the CTX level was lower than 150 pg/mL, they were

offered a drug holiday. If they declined, if the CTX level was above 150 pg/mL at baseline, or after the

drug holiday, they had extractions performed under local anesthesia. Age-matched controls not on bisphosphonates

were identified.

Results: Nine hundred fifty patients had 2,461 extractions. One hundred eighty-one patients had a CTX

level lower than 150 pg/mL. Four patients developed BRONJ; all had a CTX level lower than 150 pg/mL. All

were on alendronate. The case–control comparison approached significance (<150 pg/mL; P = .073).

Alendronate was associated with a low CTX level (P < .05). A CTX level lower than 150 pg/mL had a sensitivity

of 100% and specificity of 81%. Bayesian analysis yielded a population expected risk of BRONJ of

0.29% (95% confidence interval, 0.12-0.52); the expected risk was 0.42% for a CTX level lower than

150 pg/mL and 0.13% for a CTX level higher than 150 pg/mL.

Conclusion: The risk of BRONJ for patients with osteoporosis on bisphosphonates having extractions is

approximately 0.2%. A CTX level lower than 150 pg/mL is sensitive and is associated with an approximately

3-fold greater risk of BRONJ.

2014 American Association of Oral and Maxillofacial Surgeons

J Oral Maxillofac Surg 72:1456-1462, 2014

Author Response

Dear Reviewer,

Thank you for the feedback.  The following changes have been made accordingly.

The first paragraph has been revised and the references for the term MRONJ and DRONJ have been added.

English grammar and punctation have been further revised.

Evidence on the increase incidence of ONJ with tooth extractions has been further added, and it being a risk factor of MRONJ has been inserted in the summary.

The paper about CTx as a risk predictor tool for MRONJ has been added as reference in the summary.  Though the paper has clinical implication, as it is a rather new finding and the use of bone turnover markers for osteoporosis in clinical practice has been unclear and not widely adopted, the detail of the paper thus has not been discussed as it may extend out of the scope of current review focus.

The current recommendation of drug holiday after 3-5yrs' bisphosphonate treatment has been added in the last paragraph.

Thank you for your consideration.

Yours sincerely

Round 3

Reviewer 1 Report

Many thanks for your consideration

Author Response

Manuscript has been revised.  Thank you for your consideration.

Reviewer 2 Report

Great improvements provided, thank you